# Can ChatGPT Perform Reasoning Using the IRAC Method in Analyzing Legal Scenarios Like a Lawyer?

**Xiaoxi Kang[1], Lizhen Qu[2*], Lay-Ki Soon[1], Adnan Trakic [2]**
**Terry Yue Zhuo[2], Patrick Charles Emerton[3], Genevieve Grant[2]**
School of Information Technology, Monash University Malaysia[1],
Monash University[2], Deakin University[3]
[1]{xiaoxi.kang,soon.layKi } @monash.edu
[2]{lizhen.qu,adnan.trakic,terry.zhuo,genevieve.grant} @monash.edu
[3] p.emerton@deakin.edu.au

## Abstract

Large Language Models (LLMs), such as CHATGPT, have drawn a lot of attentions recently in the legal domain due to its emergent ability to tackle a variety of legal tasks. However, it is still unknown if LLMs are able to analyze a legal case and perform reasoning in the same manner as lawyers. Therefore, we constructed a novel corpus consisting of scenarios pertain to Contract Acts Malaysia and Australian Social Act for Dependent Child. CHAT-GPT is applied to perform analysis on the corpus using the IRAC method, which is a framework widely used by legal professionals for organizing legal analysis. Each scenario in the corpus is annotated with a complete IRAC analysis in a semi-structured format so that both machines and legal professionals are able to interpret and understand the annotations. In addition, we conducted the first empirical assessment of CHATGPT for IRAC analysis in order to understand how well it aligns with the analysis of legal professionals. Our experimental results shed lights on possible future research directions to improve alignments between LLMs and legal experts in terms of legal reasoning.

## 1 Introduction

Since CHATGPT was released by OpenAI in November 2022, there are fast-growing interests in applying LLMs to analyzing legal documents (Katz et al., 2023) and reasoning tasks (Huang and Chang, 2022). Although the recently released LLMs demonstrate strong abilities to solve challenging tasks requiring reasoning, people find that they often follow different, or even wrong reasoning paths to obtain correct answers (Paul et al., 2023; Tang et al., 2023). This issue is also referred to as a *misalignment* problem between LLMs and humans. As this problem has not been investigated in the legal domain, this work focuses on

understanding to what degree CHATGPT is able to perform reasoning for legal scenario analysis in the same way as legal professionals. Herein, we chose IRAC (Alsagoff, 1996), standing for Issue, Rule, Application, and Conclusion, as the framework for legal analysis because it is the most popular analysis methodology used by legal professionals and law schools.

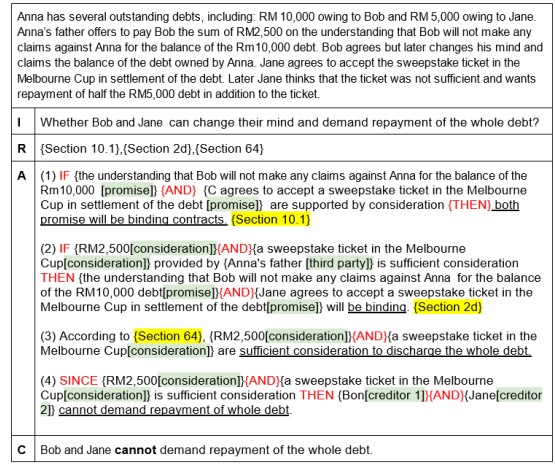

Figure 1: An example IRAC analysis conducted by a legal professional.

A typical example of IRAC analysis is depicted in Fig. 1. Given a scenario regarding the Contract Act Malaysia (CAM), the task is to answer the legal question recognized as an issue in I. In this example, the rules (R) are the relevant statutes in CAM. In a common law system, rules may also include precedents. The analysis or application (A) consists of the reasoning paths leading to the answer in the conclusion (C). Herein, a reasoning path is a sequence of reasoning steps, where each step involves taking a statement from the available facts in a given scenario and then applying a relevant rule to it. Section 2 describes IRAC in details and its importance in legal reasoning.

Interpretability of model outputs is crucial for legal professionals in real-world legal applications

---

*Corresponding author.

(Wu et al., 2022). For scenario analysis using IRAC, it is helpful for models to produce individual reasoning paths and their associated rules or precedents so that legal professionals are able to understand why models draw certain conclusions. Moreover, it is noteworthy that legal reasoning is *defeasible* such that conclusions can be overturned by considering new evidence or making different assumptions due to missing information (Sartor, 1995). However, prior legal datasets do not include intermediate reasoning paths understandable by legal professionals and neglect the aspect of defeasible reasoning.

To address the above issues, we constructed the *first* semi-structured IRAC corpus, coined SIRAC[1], which includes a set of legal scenarios pertinent to CAM and Australian Social ACT (ASA) respectively. To make IRAC analysis understandable by both LLMs and legal professionals, we proposed a semi-structured language and used it to codify an IRAC analysis for each scenario. As all scenarios are analyzed by senior law students with IRAC, we conducted detailed comparative studies between their results and the ones produced by CHATGPT, by applying different prompting and in-context learning techniques (Dong et al., 2022). As our scenarios are complex and involve 7.05 reasoning paths on average, we also decomposed the legal question of a scenario into multiple simpler questions and instructed CHATGPT to address them separately. As a result, we obtained the following novel findings via extensive analysis conducted by the law students:

- Without IRAC analysis from legal professionals, CHATGPT achieves an F1 of 0.49 on average for answering legal questions of scenarios. However, CHATGPT fails to produce complete and correct reasoning paths toward answers for any evaluated scenario albeit some of the answers are correct.

- We demonstrated the importance of providing correct intermediate reasoning paths to CHATGPT. The average F1 score of the final answers estimated by CHATGPT was improved more than 0.86, when the complete human-written reasoning paths except final answers are fed to the model.

- CHATGPT benefits from adding similar example scenarios with IRAC analysis during in-context learning, only if similar scenarios can be found.

- Decomposing legal questions into simpler ones consistently improve the accuracy of identifying legal concepts, such as "invitation to treat". However, this approach does not always improve the correctness of produced reasoning paths.

## 2 Background

A legal problem could be as simple as ascertaining the amount of money due from a tenant to the landlord under a contract affected by a certain unseen event like COVID-19, or a more complex one involving contracts between large corporate entities negotiating cross-border sales and related matters. Legal problems require unique reasoning skills to be solved. These unique skills are applied for solving legal problems in a rather systematic manner using the IRAC methodology (Alsagoff, 1996).

Before one begins to solve a legal problem using the IRAC's legal reasoning process, it is essential to acquire factual details from the client. These facts will lead to the correct identification of the legal issue. Given the identified issue, the next step is to determine the law that governs that aspect of the legal problem. The application of the law or the analysis in the third stage is perhaps the most detailed stage of IRAC. In this stage, the law is practically applied to the facts and the issues in question. As no two legal problems would be identical, one needs to be aware of possible variations. This stage is particularly important because it is here that legal reasoning skills are truly tested. The final stage of IRAC is the conclusion, which is a natural outcome of applying the law to the facts and issues. However, applying the same law to the same facts and issues could result in different conclusions, and the correct answer depends on the application of the appropriate law to the facts, which could be interpreted differently.

## 3 Dataset Construction

We constructed SIRAC to evaluate to what degree LLMs and the other AI models are able to conduct IRAC analysis in the same manner as legal professionals. Our goals are to promote research on devising novel legal AI models that are i) equipped

---

[1]Github link:
https://github.com/christinakang/SIRAC.git

with strong defeasible reasoning capability, and ii) interpretable by both legal and computer science professionals. We start with selecting scenarios pertaining to CAM and ASA, followed by conducting full-fledged IRAC analysis on all scenarios using a proposed semi-structured language. The data quality is ensured by our strict procedures.

## 3.1 Selection of Scenarios

We have selected CAM and ASA scenarios in the common-law systems, which are under-explored by legal AI research communities so that LLMs are less likely memorize similar scenarios, relevant statutes and precedents during pre-training. The IRAC analysis of a single legal scenario may take several hours or days, even for law professors and the best law students. At this stage, we do not consider scenarios pertaining to more than one areas of law, involving multiple topics and complex legal issues, in order to keep the analysis time within our budget.

We started with selecting a set of statutes in ASA before creating scenarios. In particular, Section 2, Section 3 and Section 4 in ASA are selected for the purpose of this study. Based on the statutes, law students were instructed to write 30 scenarios inspired by real-world cases so that i) each scenario is designed to answer a legal question "Is the child a dependent child"; ii) all relevant legal rules are covered by the selected sections. As a result, LLMs will be given complete information to perform IRAC analysis.

For the scenarios pertinent to CAM, we increased the complexity of IRAC analysis by considering scenarios with more diverse legal issues, requiring the application of both statutes and precedents - decisions made in previous similar cases. Herein, 20 scenarios were selected from tutorials, text books and examination questions related to CAM. Each scenario is associated with one topic and there are 10 topics in total. Each scenario pertains to a different yes-no question as the issue. For example, "Whether Bob and Jane can change their mind and demand repayment of the whole debt?", as shown in Fig. 1.

## 3.2 Annotation of IRAC Analysis

*Issue.* As issues have already been determined for creating the scenarios, law students focused on analyzing relevant statutes and precedents, in order to obtain correct conclusions to the issues.

*Rule.* We follow the format that is widely used by legal professionals to reference relevant statutes and prior cases. An example is depicted in Fig. 1.

*Application.* Law students were instructed to annotate complete reasoning paths toward conclusions. A reasoning path is a sequence of reasoning steps and the last step is the answer to the issue of a scenario. In addition, we borrowed the key ideas from *default logic* (Reiter, 1980) when performing defeasible reasoning. It differs from the well-known monotonic logic, such as first-order logic, largely by introducing defaults (assumptions) and default rules. Both defaults and default rules are applied when coping with missing information. For example, if the marriage status is unknown for a twenty-year-old defendant, it is reasonable to assume that he or she is single during reasoning. Default reasoning allows changes of intermediate conclusions if the new facts or evidences are included.

As a reasoning path may involve arguments and assumptions articulated in natural language, logical operators, and references to statutes or precedents, we introduce a semi-structured language to precisely describe the reasoning paths, so that they are easy-to-interpret by legal professionals and easy-to-process by computer programs. Inspired by neuro-symbolic approaches (Hitzler, 2022), we design the language by using a mix of the language used in statutes and symbols for logical operators. We take into account the language used to formulate statutes because it is unambiguous for legal professionals, and the same words were consistently used for the same concepts. The introduction of logical operators, such as AND and OR, is expected to facilitate future research on incorporating symbolic or neuro-symbolic reasoning into legal reasoning. In addition, we annotate mentions of legal concepts with brackets because they often play an important role in legal analysis.

*Conclusion.* We apply the same semi-structured language to formulate conclusions in the corresponding section of the IRAC structure. For each scenario, each conclusion is essentially the answer to the corresponding issue.

To facilitate the manual annotation work, we build an online annotation tool A.1 for law students, which supports annotations of legal concepts, rules, logical operators, and analysis using the semi-structured language in the IRAC framework.

### 3.3 Quality Assurance

To ensure the quality of the scenarios and annotations, we applied a strict procedure in selecting the annotators, as well as checking their progress.

*Annotators Selection and Training.* We recruited five senior law students from different universities after interviewing all the candidates. The recruited students were expected to be familiar with ASA and CAM. Before they joined the annotation project, we provided test cases to them, in order to get them familiar with the semi-structured language and the annotation tool. After they submitted the test cases, we asked the law professors to verify their answers before they started the actual annotation work.

*Scenario Quality.* To ensure the quality, the scenarios of CAM were selected from reliable sources, such as law textbooks and examination questions. Each scenario related to ASA was checked by another student besides the one who created it. All scenarios were double-checked by one expert in the area to ensure that they are reasonable and meet all criteria mentioned above. In case of disagreements on the criteria of the scenarios, the expert discussed with all the law students and revised the corresponding guidelines, if necessary, to ensure consistency.

*IRAC Analysis Quality.* We created detailed guidelines for performing and annotating IRAC analysis, as well as the usage of the annotation tool. We had regular meetings with the law students to check on their progress and to discuss any potential problems. The solved problems were incrementally reflected in the guidelines. In addition, we regularly assigned the same scenarios to different law students to let them work independently, such that we can calculate inter-annotator agreement (IAA) and use the IAA scores to check the consistency of their annotations. Moreover, the answers to the issues of the CAM scenarios were checked against the sample answers provided by the textbooks, examination questions and tutorials, so that the correctness is guaranteed.

### 3.4 Data Statistics

*Basic Statistics.* We have 50 annotated legal scenarios in total. Table 1 summarizes the statistics of the dataset. For CAM and ASA, the rules include references to precedents. In the reasoning paths, each reasoning steps is the statement from the facts of the scenario and applied with the rule. The average length of the reasoning paths is 7.05. For CAM the

average reasoning path is 9.3 because the scenario is more complex compare to ASA.

| | Scenarios | Issues | Rules | Ave Length |
|---|---|---|---|---|
| SIRAC_ASA | 30 | 1 | 3 | 4.8 |
| SIRAC_CAM | 20 | 20 | 55 | 9.3 |
| SIRAC | 50 | 21 | 58 | 7.05 |

Table 1: Basic statistics of SIRAC.

*Comparison with Related Datasets.* We compare SIRAC with the current datasets for legal QA tasks along six dimensions: i) if the tasks require legal reasoning; ii) if the data is easy-to-process by machines; iii) if the reasoning paths are intepretable by legal professionals; iv) if IRAC analysis is applied; v) if there are complete and detailed reasoning paths annotated; vi) if the reasoning requires both statutes and prior cases. As summarized in Table 2, LEGALBENCH (Guha et al., 2022) is the only one applying the IRAC methodology. However, they do not perform full IRAC analysis on scenarios. Hence, the corresponding reasoning paths are incomplete. The length of the annotated paths are also fairly short, which contain less than three steps. The other similar dataset is SARA (Holzenberger and Van Durme, 2021), which contains both question answering tasks and reasoning paths. However, they employed Prolog to codify reasoning paths, which is rather challenging for legal professionals to understand.

## 4 Assessment of ChatGPT

Our preliminary results show that CHATGPT is one of the best performing LLMs in legal reasoning. Therefore, it was chosen to assess its ability in performing IRAC analysis. Instead of evaluating only the final results of statutory reasoning by GPT3 (Blair-Stanek et al., 2023a), we employed CHATGPT to perform full IRAC analysis, and the quality of its outputs was assessed by the law students in the respective sections: Rule, Application, and Conclusion. Issues were not evaluated because they are part of the inputs. Moreover, we also carried out in-context learning and question decomposition, which have been found useful in other reasoning tasks (Mialon et al., 2023), to improve the performance of CHATGPT.

### 4.1 Evaluation Measures

In this work, we focus on understanding the pros and cons of CHATGPT on this task, as well as

| Dataset name | Reasoning | Friendly for AI systems | Friendly for legal professionals? | IRAC applied ? | Detailed reasoning paths | Analysis requiring statutes and precedents |
|---|---|---|---|---|---|---|
| SARA (Holzenberger and Van Durme, 2021) | Yes | Yes | No | No | No | No |
| COLIEE (Rabelo et al., 2022) | Yes | Yes | No | No | No | No |
| CUAD (Hendrycks et al., 2021) | No | Yes | No | No | No | No |
| MAUD (Wang et al., 2023) | No | Yes | Maybe | No | No | No |
| LEGAL BENCH (Guha et al., 2022) | Yes | Maybe | Maybe | Yes | No | No |
| Semi-structured IRAC | Yes | Yes | Yes | Yes | Yes | Yes |

Table 2: Comparison of the datasets for legal QA and legal scenario analysis.

its alignment to the reasoning behavior of humans. Therefore, the human evaluation measures are based on the marking rubrics used by law schools, and the ones for evaluating language models. Throughout all experiments of human evaluation, we formulate the majority of the measures as a statement and score each of them with either -1, 0 or 1, indicating *disagree*, *neutral*, and *agree* respectively. The statements are formulated in the way that the higher scores the better. All measures are assessed first for each selected scenario and their results are summarized for later analysis. Appendix A.3.2 shows the questions we asked the annotators.

**Rule.** The rules and precedents are evaluated with regard to *information relevance*. The corresponding statement is formulated as "The referenced statutes and precedents are correct and complete.".

**Application.** We apply multiple measures to assess the legal analysis in this section.

*Correctness of Concept Identification.* This measures whether all important legal concepts correctly identified, and is formulated as "All key concepts are correctly identified."

*Articulation of Reasoning.* This is concerns with articulation of reasoning steps and if the logical structures between steps are correct. The corresponding statement is "CHATGPT performs and articulates reasoning correctly and consistently."

*Evaluation of Assumptions.* As legal reasoning is defeasible, we evaluate the generated assumptions by using the following questions:

- How many assumptions are made?

- Among the assumptions made by CHATGPT, how many assumptions are correct?

- Compare the assumptions from CHATGPT and humans, how many of them are matched?

*General Comment.* We also ask annotators to comment on the reasoning performance of CHATGPT, in particular its strengths and weaknesses.

**Conclusion.** *Correctness* is a commonly evaluated feature (Hämäläinen and Alnajjar, 2021). Humans will evaluate the answer to identify whether it addresses the question and whether it is consistent with the source material.

For all IRAC analysis, we also evaluate the *fluency* of the texts generated by CHATGPT, in order to assess if they are understandable by humans.

To evaluate the quality of the assessment results, we reassigned 18 finished scenarios to three different annotators and asked them to evaluate the scenarios in terms of the above measures. We choose Cohen's Kappa score (Cohen, 1960; Zhan et al., 2023) to compute the agreement. The Kappa score for all the evaluation measures is 0.55, while the Kappa score for the analysis without the assumption evaluation is 0.75. Our investigation shows that the questions for the assumptions are subjective for law students, which however is a common problem in law education, as confirmed by a lecturer teaching law for several years.

### 4.2 Results and Discussions

We started with evaluating the final answers generated by CHATGPT, followed by analyzing its reasoning steps and references.

**Evaluation of Conclusion.** We fed a scenario and its issue as the input to CHATGPT. As the questions in all scenarios are all yes-no questions, we appended "Answer in one word." to the last line of the inputs so that we could automatically extract the answers from CHATGPT. An example input is depicted in Fig. 2. In addition to the scenarios pertaining to ASA and CAM, we included the 276 scenarios pertinent to the Internal Revenue Code (IRC) from the SARA dataset, where IRC is the

domestic portion of federal tax law in the United States.

Figure 2: An example prompt for answer evaluation.

|        | **Precision** | **Recall** | **F1** |
|--------|---------------|------------|--------|
| IRC    | 0.29          | 0.43       | 0.35   |
| ASA    | 0.75          | 0.60       | 0.67   |
| CAM    | 0.50          | 0.40       | 0.44   |
| Average| 0.51          | 0.48       | 0.49   |

Table 3: Result of answers produced by CHATGPT.

As shown in Table 3, CHATGPT's performance varied across these domains, with the highest precision in the ASA (0.75), and the lowest in IRC (0.29). We also compared F1 score is lowest on the IRC which is 0.35 and highest 0.67 one ASA. The average F1 score of CHATGPT is 0.49. Which we can see there are still a lot of room for improvement in the following steps.

**Evaluation of Application and Rule.** Apart from conclusions, we evaluated the ability of CHAT-GPT by letting it produce intermediate reasoning paths. It can be achieved by removing "Answer in one word." from the last line. As SARA does not contain human annotated reasoning paths, we evaluated the sections of Application only on the scenarios of CAM and ASA.

We selected 20 scenarios per area for human evaluation. Among the 40 scenarios we evaluated, only two of them produced high quality reasoning paths, which are considered as *agreed* in the question about *articulation of reasoning*. Further details can be found in the example depicted in Fig 3.

Table 5 summarizes CHATGPT's performance in CAM and ASA in terms of all evaluation measures, including fluency, information relevance, articulation of reasoning. From the results we can see that CHATGPT receives much less *agree* (1) than that evaluated on the answers. In contrast, annotators agree that the fluency of CHATGPT outputs is high, in line with the observations from related work. However, CHATGPT does not provide enough information such as references to statutes

Figure 3: An example reasoning path.

or precedents in the involved reasoning paths. Out of 40 examples, there is only one example from SIRAC-CAM having the analysis with correct references to statutes and precedents. The performance is also poor on the articulation of reasoning. We notice that the formulation of the analysis from CHATGPT is sometimes very confusing and logically inconsistent.

The evaluation results of assumptions are listed in Table 4. We evaluated the result into two different perspectives: assumption and analysis. The assumption measures if the answer from CHATGPT makes reasonable assumptions. For example: if my son stay at my home whole day and doing nothing. Then we can assume that my son is wholly dependent on me. However, we also noticed that, although some of the assumptions were identified by CHATGPT, they were not analysed correctly. For example: CHATGPT answered "my son is **not** wholly dependent on me", which is a wrong assumption. We compared the result from different law and different methods. The result indicate that, after using the in-context learning and decomposed questions, more assumptions were identified and discussed correctly. Nonetheless, the improvement of analysis still lesser than the assumptions.

Overall, although CHATGPT could produce correct conclusions in IRAC analysis, its analysis part in Application mostly are not aligned with those from legal professionals. The references to law and

precedents are often missing or incorrect.

| | Assumptions | | | Analysis | | |
|---|---|---|---|---|---|---|
| | Precision | Recall | F1 Score | Precision | Recall | F1 Score |
| Overall | 0.45 | 0.68 | 0.54 | 0.28 | 0.43 | 0.34 |
| CAM | 0.56 | 0.66 | 0.61 | 0.35 | 0.41 | 0.38 |
| ASA | 0.32 | 0.73 | 0.45 | 0.20 | 0.46 | 0.28 |
| In-context Overall | 0.73 | 0.96 | 0.83 | 0.58 | 0.76 | 0.66 |
| In-context CAM | 0.54 | 0.95 | 0.69 | 0.32 | 0.57 | 0.41 |
| In-context ASA | 0.94 | 0.97 | 0.96 | 0.85 | 0.88 | 0.87 |
| Decomposition Overall | 0.75 | 0.88 | 0.81 | 0.48 | 0.57 | 0.52 |
| Decomposition CAM | 0.88 | 0.90 | 0.89 | 0.49 | 0.50 | 0.49 |
| Decomposition ASA | 0.66 | 0.87 | 0.75 | 0.48 | 0.63 | 0.54 |

Table 4: Performance of CHATGPT in terms of Precision, Recall, and F1 Score.

**Impact of Adding Reasoning Paths.** LLMs are known to rely on "shortcuts" for predictions occasionally (Du et al., 2022), hence their robustness is questionable. In this experiment, we verify if CHATGPT's performance can be improved by progressively providing it with human-authored parts of reason paths, in order to shed light on future research directions of legal reasoning. Herein, we add 20%, 40%, and 80% of the reasoning paths annotated by law students to the inputs after the legal question. The final outcome has been progressively improved. Figure 4 shows that the more analysis we provided in a prompt, the higher F1 scores we gained from the final answers. The F1-score is able to reach 0.89 /1.0 starting from the lowest 0.10/0.0 for CAM/ASA respectively. This observation suggests that LLMs, e.g. CHATGPT, should focus on generating correct intermediate steps. We also found out that this observation is consistent in both areas of law that we evaluated.

**Effects of Adding Examples to Prompts.** To understand the performance of CHATGPT using in-context learning, we added the most similar example in terms of topics or related statutes to the prompt for each scenario. Fig 5 gives an example of in-context learning.

We evaluated the same 40 scenarios as before. The quality of the reasoning paths is improved by 27.5%, especially for ASA, because the scenarios in this area are similar to each other. From the

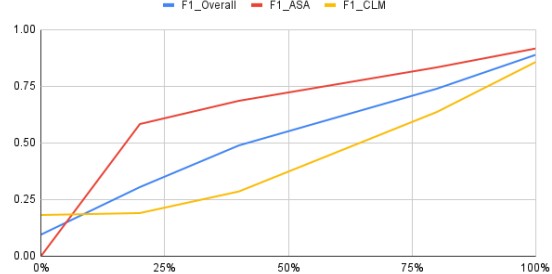

Figure 4: Result of the add reasoning paths.

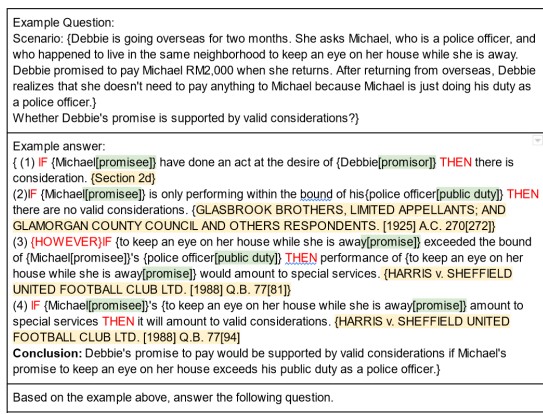

Figure 5: An example of adding Examples to Prompts.

statistics of our data, we can tell that all scenarios are related to one topic and two sections pertinent to ASA. The followings are the main findings according to our human evaluation:

- The analysis parts are improved in 50% of the given scenarios after adding examples.

- The analysis parts are improved for the scenarios that are similar to the provided examples.

Table 4 displays the F1 score, changing from 0.34 to 0.66 on the analysis part. The ASA is able to cover more legal points and F1 scores are improved up to 0.96. Referring to the example provided, CHATGPT is able to answer similar to the one shown in the example question.

**Effects of Decomposed Questions.** If CHATGPT is unable to capture the main legal concepts

| | | Correctness | Fluency | Information relevance | Concept Identification | Articulation Reasoning |
|---|---|---|---|---|---|---|
| ASA | -1 | 8 | 0 | 20 | 9 | 13 |
| | 0 | 12 | 1 | 0 | 11 | 7 |
| | 1 | 0 | 19 | 0 | 0 | 0 |
| CAM | -1 | 8 | 1 | 19 | 8 | 8 |
| | 0 | 10 | 1 | 0 | 11 | 10 |
| | 1 | 2 | 18 | 1 | 1 | 2 |
| Total | -1 | 16 | 1 | 39 | 17 | 21 |
| | 0 | 22 | 2 | 0 | 22 | 17 |
| | 1 | 2 | 37 | 1 | 1 | 2 |

Table 5: Result for reasoning paths.

| | | Correctness | Fluency | Improvement | Information relevance | Concept Identification | Articulation Reasoning |
|---|---|---|---|---|---|---|---|
| ASA | -1 | 5 | 0 | 3 | 5 | 5 | 5 |
| | 0 | 7 | 3 | 2 | 9 | 6 | 5 |
| | 1 | 8 | 17 | 15 | 6 | 9 | 10 |
| CLM | -1 | 11 | 0 | 9 | 11 | 10 | 11 |
| | 0 | 6 | 3 | 7 | 6 | 7 | 4 |
| | 1 | 3 | 17 | 4 | 3 | 2 | 5 |
| Total | -1 | 16 | 0 | 12 | 16 | 15 | 16 |
| | 0 | 13 | 6 | 9 | 15 | 13 | 9 |
| | 1 | 11 | 34 | 19 | 9 | 11 | 15 |

Table 6: Result for adding examples to prompts.

| | | Correctness | Improvement | Fluency | Information relevance | Concept Identification | Articulation Reasoning |
|---|---|---|---|---|---|---|---|
| ASA | -1 | 10 | 12 | 0 | 20 | 10 | 9 |
| | 0 | 7 | 5 | 0 | 0 | 7 | 10 |
| | 1 | 3 | 3 | 20 | 0 | 3 | 1 |
| CAM | -1 | 10 | 11 | 0 | 18 | 9 | 12 |
| | 0 | 6 | 4 | 3 | 2 | 7 | 5 |
| | 1 | 4 | 5 | 17 | 0 | 4 | 3 |
| total | -1 | 20 | 23 | 0 | 38 | 19 | 21 |
| | 0 | 13 | 9 | 3 | 2 | 14 | 15 |
| | 1 | 7 | 8 | 37 | 0 | 7 | 4 |

Table 7: Result for decomposed questions.

for the reasoning analysis, we investigated how it can be improved by decomposing a question into smaller questions? Inspired by Chain-Of-Thoughts (Wei et al., 2022), we attempted to dissect the issue questions into smaller questions and therefore guided CHATGPT towards more accurate answers. Fig 6 shows examples of decomposed questions. The list of all decomposed questions can be referred to in appendix A.3.1. Our annotators decomposed the issue questions based on the legal concepts that need to be mentioned in the reasoning paths.

From the sample results, we can observe that with decomposed questions, CHATGPT is able to apply the analysis to the facts identified from the scenario, followed by matching and aligning them with the legal concepts.

```
Decomposed Questions:
5.1. Is the promise supported by considerations?
5.2. If it is supported by considerations then whether the considerations are valid?
5.3 If considerations are valid then whether they would be bound by the promise?
5.4 If they are bound by the promise, will Bob and Jane change their mind and demand repayment
of the whole debt?

ChatGPT Answer:
5.1. Is the promise supported by considerations?
Yes, Y's promise to pay Bob and Jane in exchange for the release of Anna's debts is supported by
considerations. (Correct Answer)
5.2. If it is supported by considerations then whether the considerations are valid?
The consideration provided by Anna to Bob and Jane is valid, as it involves the payment of money
and the provision of a sweepstake ticket. (Neutral, the consideration is correct, but lacks facts and
justification.)
5.3 If considerations are valid then whether they would be bound by the promise?
Bob would be bound by the promise made by Anna, as he agreed to accept the payment in
exchange for releasing Anna from the remaining debt. Jane would also be bound by the promise,
as he accepted the sweepstake ticket as settlement for the debt owed by Anna. (Correct Answer)
5.4 If they are bound by the promise, will Bob and Jane change their mind and demand repayment
of the whole debt?
Bob cannot change his mind and demand repayment of the remaining debt, as he has already
agreed to accept the payment from Anna in exchange for releasing X from the remaining debt.
Jane cannot demand repayment of half the debt in addition to the sweepstake ticket, as he has
already accepted the ticket as settlement for the debt owed by Anna. Therefore, both Bob and
Janeare bound by the promise made by Anna. (Correct Answer)
```

Figure 6: Example of Decomposed questions.

Table 7 shows the result of the decomposed questions. We can see improvement from previous answers. From Table 4, the overall legal concept identification maintains high performance with a precision of 0.75, recall of 0.88, and an F1-score of 0.81. The results prove that decomposed questions help in identifying legal concepts related to the scenario more effectively.

# 5 Related Work

In this section, we present related work, highlighting the similarities as well as the differences between theirs and ours. Previous work related to our paper can be categorized into the following subsections.

## 5.1 LegalQA with Reasoning Task

There are several research related to legal reasoning. Nils (Holzenberger and Van Durme, 2021) presented the reasoning step with Prolog (De Raedt et al., 2007). Their research focused on identifying the rules applicable in the scenarios. They also attempted to do the reasoning by rewriting the rule with Prolog (De Raedt et al., 2007). However, it is not applicable to other law and Prolog is not easily comprehensible by legal professionals. (Yu et al., 2022) proved that the best accuracy can be obtained by applying the IRAC methodology. Their work did not evaluate the reasoning path, which is the analysis part of IRAC. The accuracy was only based on the conclusion, which is not a comprehensive evaluation of IRAC methodology. Guha (Guha et al., 2022) separated IRAC to four individual tasks. They pplied different data corpus and fit into IRAC analysis. However, they did not apply the complete IRAC on a single case. In comparison to all these related works, our work presented in this paper covers the complete IRAC methodology. More importantly, the reasoning traces are presented in a comprehensible method, for both legal and IT professionals.

## 5.2 Reviewing CHATGPT Performance

Given the proliferation of CHATGPT since November 2022, a lot of research works were carried out to review the performance of the CHATGPT in specific domains (Zhuo et al., 2023). Tiffany (Kung et al., 2022) found that CHATGPT was able to pass all the three examinations without any specialized training or reinforcement. In addition, CHATGPT provided a high level of concordance and insight in its explanations.Their work paves the way for domain-specific evaluations on CHATGPT. Andrew (Blair-Stanek et al., 2023b) and Savel (Savelka, 2023) reviewed CHATGPT performance based on different input prompts. From their results, GPT-3 had 78% accuracy, raising doubts about GPT-3's ability to handle basic legal work. Similar to (Guha et al., 2022), there is no evaluation on the reasoning traces. Although it was mentioned

that CHATGPT performs poorly, no specific measures were used to support the claim objectively.

### 5.3 Alignment Problems between Large Language Models (LLMs) and Humans

Large language models (LLMs) have achieved success at a range of tasks such as question answering, summarisation, and dialogue. Hannah (Kirk et al., 2023) proposed a three-tiered policy framework that allows users to experience the benefits of personalised alignment. Similarly, Jakob (Mökander et al., 2023) also used a three-layered approach for the LLMs which include governance audits, model audits, and application audits. Atoosa (Kasirzadeh and Gabriel, 2023) developed a philosophical analysis of building blocks of linguistic communication between conversational agents and human interlocutors. Bakker (Bakker et al., 2022) studied LLMs alignment from different perspectives, their work highlights that there is still a gap between the alignment of LLMs and humans. However, there is a need to strike the balance of providing more information to LLMs and not burdening the human experts involved unnecessarily. This observation inspires us to investigate the right amount of information LLMs require from humans in order to achieve satisfactory performance.

## 6 Conclusion

We constructed a novel dataset, coined SIRAC, to evaluate the ability of LLMs for IRAC analysis. SIRAC contains 50 legal scenarios pertinent to ASA and CAM. Every scenario was annotated with reasoning paths with an average length of 7.05, described by our proposed semi-structured language. SIRAC is not only useful for the studies and analysis by legal professionals, but offers a fertile pool of resources for legal NLP.

Our evaluation results on CHATGPT show that the powerful LLMs can produce reasonable answers but mostly fail to yield correct reasoning paths aligned with legal experts. Its performance can be further improved by providing parts of the annotated reasoning paths, including similar annotated scenarios for in-context learning and decomposing complex issues into simpler questions. As those techniques are not specific to the investigated areas of law, it is desirable to understand the extend to which such empirical findings still hold in the other legal areas in the future work.

## 7 Limitations

While SIRAC offers a comprehensive resource for IRAC analysis, both by legal professionals and machine learning models, we need to acknowledge the existing constraints.

**Lack of legal domain** It is challenging to engage legal professionals who can understand the law of different countries. Hence, there is a limit to the extent of analysis that could be performed on some published legal dataset. At this stage, SIRAC is limited to only two types of law ASA and CAM. However, the methodology proposed in this paper is applicable to different laws. SIRAC covers two different laws from different countries. In the future, we plan to engage more legal professional who can contribute to expanding the dataset to other types of law.

**Lack of data resources for CHATGPT revision** due to the limited resources, we were able to engage a small group of annotators to assist us in evaluating the outcome produced by CHATGPT. While the analysis is sufficient for us to obtain concrete insights, we hope we to engage more annotators for further strengthen our research contribution.

## 8 Ethics

All the tasks carried out in this paper aim to assess the reasoning traces for the legal scenario analysis by CHATGPT. The entities mentioned in the legal scenarios are anonymized by the annotators. The court cases include in our dataset does not reveal the real case. We only include the court case name and responding paragraph numbers. In fact, court cases are accessible by the public and often used for further analysis by the commentators or for law studies. As such, the court cases used do not require additional ethics approval.

## 9 Acknowledgements

This research is made possible through the generous support of Monash Inter-Faculty Seeding Grant. We wish to extend our sincere gratitude to all the dedicated annotators who have made invaluable contributions to this project. We would like to extend our special thanks to Dr. Sia Chin Chin from Taylor's University, whose provision of the examination questions as a reliable source for the legal scenarios, along with assistance in recruiting qualified annotators, has been instrumental in advancing our research endeavors.

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

# A  Appendix

## A.1  Data Annotation Guidelines

We conduct the data annotation workshops with the annotators on a weekly basis meeting. The annotators are trained with the expert. We also devleoped a data annotation platform for the annotators. We provided training for the

# 1. Introduction

This data annotation is working on the contract law-related scenario. The general task will include legal concept identification for the given scenario and use IRAC(Issues, Rules, Analysis, Conclusion) to process the scenarios.

The second part of the annotation is the evaluation of the chatGPT's answer for the same input legal scenarios.

# 2. Scenario

We are expecting to have 50 legal scenarios related to contract law and Australia social act.

The source of the scenario is from exam questions, tutorial questions and books.

The annotator is expecting to highlight the scenario with the legal concept. You can choose the legal concept in the given list or add a new legal concept you may think is useful in this scenario.

# 3. IRAC analysis

For the IRAC analysis, there will be an IRAC form for you to fill out. The following are the details you need to fill in.
Issue: Questions and problems for the given scenario. Suggest starting the question with whether.
Section related: select sections related to the given issue
Related court case: Enter the related court case for the given issue
Analysis: It is a free text area. You can drag the predefined relations such as IF…THEN… and the legal concept you highlighted in the scenario to write the analysis in a point form. To avoid ambiguity and misunderstanding of the syntax. It is suggested to use standard legal words all the time.
Conclusion: Answer the given issues.

# 4. Annotation Platform Example

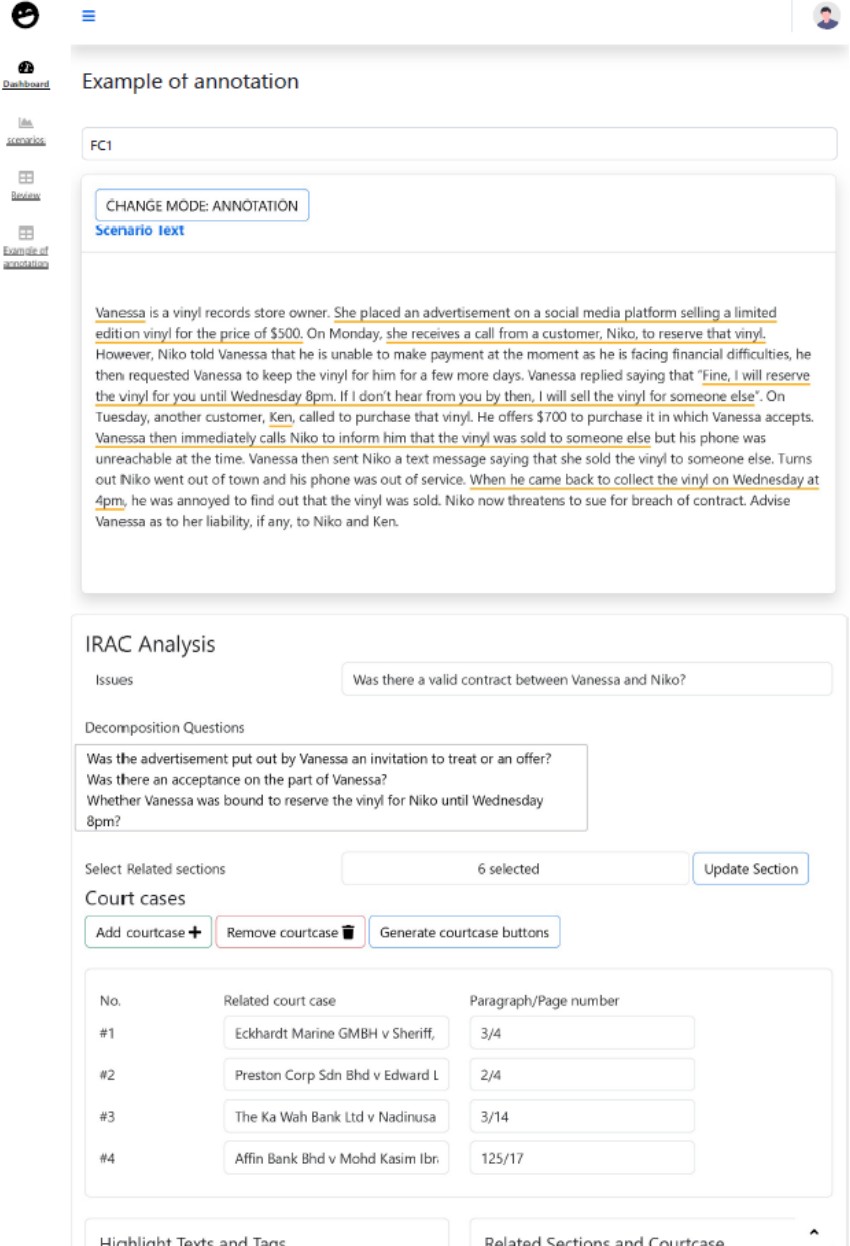

Example of annotation

FC1

CHANGE MODE: ANNOTATION

**Scenario Text**

Vanessa is a vinyl records store owner. She placed an advertisement on a social media platform selling a limited edition vinyl for the price of $500. On Monday, she receives a call from a customer, Niko, to reserve that vinyl. However, Niko told Vanessa that he is unable to make payment at the moment as he is facing financial difficulties, he then requested Vanessa to keep the vinyl for him for a few more days. Vanessa replied saying that "Fine, I will reserve the vinyl for you until Wednesday 8pm. If I don't hear from you by then, I will sell the vinyl for someone else". On Tuesday, another customer, Ken, called to purchase that vinyl. He offers $700 to purchase it in which Vanessa accepts. Vanessa then immediately calls Niko to inform him that the vinyl was sold to someone else but his phone was unreachable at the time. Vanessa then sent Niko a text message saying that she sold the vinyl to someone else. Turns out Niko went out of town and his phone was out of service. When he came back to collect the vinyl on Wednesday at 4pm, he was annoyed to find out that the vinyl was sold. Niko now threatens to sue for breach of contract. Advise Vanessa as to her liability, if any, to Niko and Ken.

## IRAC Analysis

Issues                    Was there a valid contract between Vanessa and Niko?

Decomposition Questions

Was the advertisement put out by Vanessa an invitation to treat or an offer?
Was there an acceptance on the part of Vanessa?
Whether Vanessa was bound to reserve the vinyl for Niko until Wednesday 8pm?

Select Related sections        6 selected        Update Section

## Court cases

Add courtcase +    Remove courtcase 🗑    Generate courtcase buttons

| No. | Related court case | Paragraph/Page number |
|-----|--------------------|-----------------------|
| #1  | Eckhardt Marine GMBH v Sheriff, | 3/4 |
| #2  | Preston Corp Sdn Bhd v Edward L | 2/4 |
| #3  | The Ka Wah Bank Ltd v Nadinusa | 3/14 |
| #4  | Affin Bank Bhd v Mohd Kasim Ibr | 125/17 |

Highlight Texts and Tags                Related Sections and Courtcase

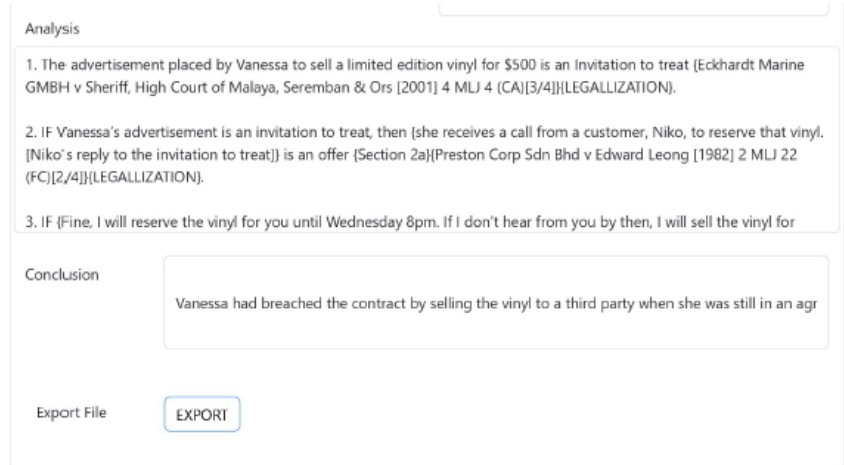

# 5. ChatGPT answer analysis

**Annotation Guidelines**

Human evaluation was conducted in the experiment. The following are the details when we go through the result of chatGPT's outcome. The annotators used the following scoring for the factors below: -1 means disagree, 0 means neutral and 1 means agree for the annotators rating the factors below.

1. **Correctness** is a commonly evaluated feature \cite{Mika2021}. Humans will evaluate the answer to identify whether it addresses the question and whether it is consistent with the source material.

First example of **Correctness**;
Human's answer:
- Since $2500 [consideration to creditor 1's promise] and a sweepstake ticket [consideration to creditor's promise] is sufficient consideration, then [creditor 1] and [creditor 2] cannot demand repayment of the whole debt.

Chat GPT's answer:
- Since $2500 [consideration to creditor 1's promise] and a sweepstake ticket [consideration to creditor's promise] is not a sufficient consideration, then [creditor 1] and [creditor 2] can demand repayment of the whole debt.

Scoring: -1, as the answer was incorrect and not matched.

Second example of **Correctness**;
Human's answer:
- Since $2500 [consideration to creditor 1's promise] and a sweepstake ticket [consideration to creditor's promise] is sufficient consideration, then [creditor 1] and [creditor 2] cannot demand repayment of the whole debt.

Chat GPT's answer:
- [creditor 1] and [creditor 2] cannot demand repayment of the whole debt.

Scoring: 0, even though the answer was correct, there is no explanation.

Third example of **Correctness**;
Human's answer:
- Since A is still in full time education, A is a dependent child.

Chat GPT's answer:
- A could be considered a dependent child as she was still in full time education.

Scoring: 1, correct with explanation.

2. **{Answerability}** Annotators are typically asked to rate the output on a fixed scale with a specific level of quality. The following are the factors that the annotators need to rate for the given answer.

a) **{Fluency}** refers to the generated text with respect to the English language such as grammar and word choice for each question. It is used to determine whether the answer is understandable for normal human beings.

First example of **Fluency**;
- Yes, that is correct. If Alice is not married and is filing a tax return for 2015, she can claim a personal exemption of $4000 under section 151(a) of the Internal Revenue Code.
- However, the actual outcome will depend on the specific terms of the contract between Jennifer and Lawrence. It is possible that the contract may have included provisions for circumstances in which Jennifer may choose to disobey Lawrence's instructions.

Scoring: 1, as the choice of words and grammar was accurate and easy to understand.

Second example of **Fluency**;
- James start (starts) work at 8am.
- Than (then), they proceed to buy the goods.
- Their (they) have accepted the offer by way of replying to the email.

Scoring: -1, grammar mistakes.

b) **{Informative Relevance}** refers to the generated text related to the source material, whether the sentence includes accurate sections of related statutes or court cases.

First example of **informative relevance**;
Chat GPT's answer:
- She can claim a personal exemption of $4000 under Section 151(a) of the Internal Revenue Code.

Scoring: 1, the sections are detailed and mentioned in the answer.

Second example of **informative relevance**;
Chat GPT's answer:
- She can claim a personal exemption of $4000 under the Internal Revenue Code.

Scoring: 0, the answer is general without referring to the specific section, it couldn't be considered as informative and relevant enough.

Third example of **informative relevance**;
Chat GPT's answer:
- She can claim a personal exemption of $4000.

Scoring: -1, the answer is too general without referring to the statute.

  c) **{Articulation of argument}** refers to the generated text demonstrating imagination or flair for the given scenario. This is to determine whether the generated text identified important legal concepts when doing the analysis.

First example of **Articulation of argument**;
- The personal exemption amount for 2015 was $4000 for individuals who are not married and are not the dependent of another taxpayer.

Scoring: 1, "Taxpayer" is the key legal concept that needs to be identified.

Second example of **Articulation of argument**;
- Since the person does provide financial support, this amounts to being in substantial care of person X. As such, the kid is a dependent child of person X.

Scoring: 1, "substantial care" is the key legal concept that needs to be identified.

Third example of **Articulation of argument**;
- Since the agreement is an agreement of apprenticeship, and that it is necessary for Jennifer's life in order for her to earn an income, then she would be liable for disobeying the instruction despite the lack of capacity.

Scoring: 1, "capacity" and "necessary for life"is the key legal concept that needs to be addressed.

**3. {Default Reasoning}**

This is the part where the annotators evaluate the reasoning performance for each generated text. We will use the precision and recall matrix to evaluate the reasoning step. Following is an example for the assumption that is highlighted by the annotators. Usually, there is limited information from the scenarios given. In order to have a good analysis for reasoning, there always should be assumptions based on the application of law.

Examples of **Default Reasoning** by annotators::
  a) Is ChatGPT making a **good assumption**? (-1,0,1)
     This is to evaluate the overall performance of the reasoning done by ChatGPT.
Scoring:
       - -1: Most assumptions made were inaccurate and not matched.
       - 0: Most assumptions made were accurate but not matched/ assumption made were inaccurate but matched
       - 1: Most assumptions made were accurate and matched.

  b) How many assumptions are made by ChatGPT? (Answer in numbers)
Examples of **an assumption**:

- In the case of B, if the agreement between B and Y was made in writing and was legally binding, then B may be bound by the agreement and may not be able to demand repayment of the balance of the debt owed by X.
- Since they are unemployed, they may not be considered a dependent child under the Australian social act.
- It is likely that she is fully dependent on you, hence, she is a dependent child.
- It could be argued that the agreement between A and B falls under this exception

Scoring: These are each considered as 1 assumption. However, it is subjective when it comes to determining the numbers of correct assumptions. (Tips: Focus on key words such as "if A, then B", "since A, then may/ may not be B", "likely that", "it could be argued that").

c) Among the assumptions made by ChatGPT, how many assumptions are **correct**? (Answer in numbers)

First example of **correct assumptions**:

Chat GPT's answer:

It is important to note that the definition of a dependent child and the criteria for determining dependency can vary between different government programs.

Scoring: 1 correct assumption, even though this was not mentioned in Human's answer, these are still relevant factors to consider. Unmatched, but correct, still a correct assumption.

Second example of **correct assumptions**:

Human's answer:
- Since the kid is dependent on X, then the kid is a dependent child of X.

Chat GPT's answer:
- I can confirm that the kid is a dependent child.

Scoring: 1 correct assumption, Regardless of whether there is reasoning/ justification/ reference to the law/ matched or not matched, so long as it is correct, then it is considered 1 correct assumption.

d) Compare the assumptions from ChatGPT and humans, how many of them are **matched**? (Answer in number)

Example of **matched assumptions**:

Human's assumption
- Since A was in an agreement of apprenticeship, then A would be liable for breaching the agreement for disobeying B's instructions.

Chat GPT's assumption
- If A failed to follow B's instruction, then she is in breach of the agreement..

Scoring: 1 matched assumption. Even though no reasoning/ justification/ reference to the law, so long as it is matched, then it is considered as 1 matched assumption.

e) Give a general comment on the performance of ChatGPT's reasoning, along with the strengths and weaknesses.

Examples for **comments**:
- Lack of discussion on the main issue.
- Too descriptive and merely restating the facts.
- Explanations were provided, legal issues were being raised and applied accurately.

Examples of **strength**:
- Good application of the law.
- Clear explanations were provided.

Examples of **weakness**:
- Lack of reference to the law.
- Irrelevant information included.
- Main issues were not mentioned.

# 6. Requirements to join the program

- The commit time for this project will be around 5-10  hours per week
- Students need to pass the pre-test before they start on the work for the formal session
- Students need to be familiar with Contract Law and Australia Social Act and get at least a B for this subject
- Data annotation will be done through face-to-face or Zoom meetings for the formal session; the venue will be Monash University.
- There is a progress update meeting every week. The expert will join the meeting to check the weekly progress.

## A.2 Number of the issues

1. Is the child dependent child?

2. Whether is there a valid contract between Alan and Cate?

3. whether there is a contract between Ellie and Frank, Frank and Galvin.

4. whether Peter can compel Trevor to pay him again

5. Whether XYZ is bound by Sam's actions.

6. whether Jennifer could be liable for breach of agreement

7. whether Jennifer could be liable for breach of agreement

8. whether B and C can change their mind and demand repayment of whole debt.

9. whether Debbie's promise is supported by valid considerations

10. whether the exemption clause is effective.

11. whether Jim could rely on exemption clause

12. whether Freedom Flight Sdn Bhd could keep the sum paid by Monash Soccer Team and whether they could claim the balance 50

13. whether Ron Realty Sdn Bhd can rely on doctrine of frustration

14. whether Cathy is legally obliged to pay Dave RM500 and RM1000 and whether Cathy is legally obliged to buy a car for Dave.

15. whether there is any enforceable contract between Lilian and Mike, whether whether there is a contract between Lilian and Neil for gold and silver dye, Whether there is an enforceable term on the quality of red dye

16. whether the contract is voidable by fraud

17. whether Dr Pritam can rescind the contract

18. whether Bakar have any contractual right against Ali and Chong.

19. what is the damages Alpha Pty Ltd is entitled to among the options of a,b,c,d and e

20. whether Southern Cross Hospital has to return the X-ray

21. whether BB have to pay compensation for breach of contract.

## A.3 Evaluation Details

**Evaluation of Assumptions.** As legal reasoning is defeasible, we evaluate the generated assumptions by using the following questions:

- Is ChatGPT making a good assumption? (-1,0,1) This is to evaluate the overall performance of the reasoning done by ChatGPT. Scoring:

  - -1: Most assumptions made were inaccurate and not matched.
  - 0: Most assumptions made were accurate but not matched/ assumption made were inaccurate but matched
  - 1: Most assumptions made were accurate and matched.

- How many assumptions are made by ChatGPT? (Answer in numbers)

- Among the assumptions made by ChatGPT, how many assumptions are correct? (Answer in numbers)

- Compare the assumptions from ChatGPT and humans, how many of them are matched? (Answer in number)

- Give a general comment on the performance of ChatGPT's reasoning, along with the strengths and weaknesses.

### A.3.1 Decomposed questions

The following is the decomposed question for each scenario.

| Law | Decomposed Questions |
|---|---|
| ASA 01 | If the child is above 16 and below 22?
If the child is wholly or substantially dependent on adult?
If the child's income does not exceed 6403 in the financial year? |
| ASA 02 | If the child is above 16 and below 22?
If the child is wholly or substantially dependent on adult?
If the child's income does not exceed 6403 in the financial year? |
| ASA 03 | If adult is legally responsible for day to day care, welfare and development of the child and the child is in the adult's
If the child is not a dependent child of someone else and is wholly and substantially in the care of the adult
If the child is in full-time education
If the child is in receipt of income which exceeds 107.70 per week |
| ASA 04 | If the child is above 16 and below 22?
If the child is wholly or substantially dependent on adult?
If the child's income does not exceed 6403 in the financial year? |
| ASA 05 | If adult is legally responsible for day to day care, welfare and development of the child and the child is in the adult's
If the child is not a dependent child of someone else and is wholly and substantially in the care of the adult
If the child is in full-time education
If the child is in receipt of income which exceeds 107.70 per week |
| ASA 06 | If the child is above 16 and below 22?
If the child is wholly or substantially dependent on adult?
If the child's income does not exceed 6403 in the financial year? |
| ASA 07 | If the child is above 16 and below 22?
If the child is wholly or substantially dependent on adult?
If the child's income does not exceed 6403 in the financial year? |
| ASA 08 | If adult is legally responsible for day to day care, welfare and development of the child and the child is in the adult's
If the child is not a dependent child of someone else and is wholly and substantially in the care of the adult
If the child is in full-time education
If the child is in receipt of income which exceeds 107.70 per week |
| ASA 09 | If the child is above 16 and below 22?
If the child is wholly or substantially dependent on adult?
If the child's income does not exceed 6403 in the financial year? |
| ASA 10 | If the child is above 16 and below 22?
If the child is wholly or substantially dependent on adult?
If the child's income does not exceed 6403 in the financial year? |
| ASA 11 | If adult is legally responsible for day to day care, welfare and development of the child and the child is in the adult's
If the child is not a dependent child of someone else and is wholly and substantially in the care of the adult
If the child is in full-time education
If the child is in receipt of income which exceeds 107.70 per week |
| ASA 12 | If the child is above 16 and below 22?
If the child is wholly or substantially dependent on adult?
If the child's income does not exceed 6403 in the financial year? |
| ASA 13 | If adult is legally responsible for day to day care, welfare and development of the child and the child is in the adult's
If the child is not a dependent child of someone else and is wholly and substantially in the care of the adult
If the child is in full-time education
If the child is in receipt of income which exceeds 107.70 per week |
| ASA 14 | If adult is legally responsible for day to day care, welfare and development of the child and the child is in the adult's
If the child is not a dependent child of someone else and is wholly and substantially in the care of the adult
If the child is in full-time education
If the child is in receipt of income which exceeds 107.70 per week |
| ASA 15 | If the child is above 16 and below 22?
If the child is wholly or substantially dependent on adult?
If the child's income does not exceed 6403 in the financial year? |

| | |
|---|---|
| ASA 16 | If the child is above 16 and below 22?
If the child is wholly or substantially dependent on adult?
If the child's income does not exceed 6403 in the financial year? |
| ASA 17 | If adult is legally responsible for day to day care, welfare and development of the child and the child is in the adult's
If the child is not a dependent child of someone else and is wholly and substantially in the care of the adult
If the child is in full-time education
If the child is in receipt of income which exceeds 107.70 per week |
| ASA 18 | If the child is above 16 and below 22?
If the child is wholly or substantially dependent on adult?
If the child's income does not exceed 6403 in the financial year? |
| ASA 19 | If the child is above 16 and below 22?
If the child is wholly or substantially dependent on adult?
If the child's income does not exceed 6403 in the financial year? |
| ASA 20 | If adult is legally responsible for day to day care, welfare and development of the child and the child is in the adult's
If the child is not a dependent child of someone else and is wholly and substantially in the care of the adult
If the child is in full-time education
If the child is in receipt of income which exceeds 107.70 per week |
| CAM 01 | 1. Was the advertisement put out by Alan an invitation to treat or an offer
2. Whether Alan has accepted Cate's offer to buy the book at 1000
3. Whether Alan is bound to keep his offer open for 7 days
4. Whether there is valid acceptance by Cate for Alan's offer to sell at 2000 |
| CAM 02 | 1. whether Ellie's advertisement is an invitation to treat or an offer
2. whether Frank has accepted Ellie's offer
3. Whether there is valid revocation of offer by Ellie
4. whether Galvin has accepted Frank's offer?
5. Could Frank communicate his revocation of offer through third party?
6. Could Galvin revoke his acceptance after sending the text? |
| CAM 03 | 3.1. Does the agent have any expressed or implied authority?
3.2. If the agent has authority, then whether the agent's action exceeds his authority?
3.3. If the agent's actions have exceeded his authority, then would the principal be bound by the agent's actions? |
| CAM 04 | 1. Does Sam have any authority?
2. Whether Sam's authority is implied or express
3. whether Sam's action of purchasing land on behalf of XYZ Bhd exceeds his authority?
4. Whether Sam has apparent authority
5. What can XYZ Bhd do if Sam has exceeded his authority? |
| CAM 05 | 1. Is Jennifer a competent party to to contract
2. If she is not, would she be bound by the agreement?
3. Is the agreement between Jennifer and Lawrence a contract for necessary? |
| CAM 06 | 1. Is Jennifer a competent party to to contract
2. If she is not, would she be bound by the agreement?
3. Is the agreement between Jennifer and Lawrence a contract for necessary? |
| CAM 07 | 1. Is B's promise to forgo the rest of the debt supported by considerations?
2. Does the part payment of debt by X's father amount to valid considerations?
3. Is C's promise to forgot the debt supported by considerations?
4. Does the sweepstake ticket amount to valid considerations? |
| CAM 08 | 1. Does Michael's action amount to valid considerations?
2. Does Michael's action amount to valid considerations despite his public duty as a police officer? |
| CAM 09 | 1. whether the exemption clause on the ticker is incorporated into the contract
2. Does the small print on the reverse side of the ticket amount to sufficient notice?
3. whether the exemption clause on the sign is incorporated into the contract
4. if the exemption clause is incorporated, would it be effective?
5. how would the court interpret the exemption clause
6. Would the exemption clause be effective if the stain marks are caused by negligence?
7. What can Annie do if the exemption clause is incorporated ? |

| | |
|---|---|
| CAM 10 | 1. is the exemption clause on the docket incorporated into the contract?
2. Is the exemption clause on the docker incorporated by past dealings?
3. how would the court interpret the exemption clause if it is incorporated into the contract?
4.  Would Jim's actions of deviation from direct route affect the effectiveness of the exemption clause?
5. Does Jim's actions of deviation from direct route amount to fundamental breach? |
| CAM 11 | 1. whether the contract is frustrated by air traffic controllers' strike
2. Does the air traffic controllers' strike render the contract impossible
3. What can the Monash Soccer Team do if the contract is frustrated?
4. What can Freedom Flight Sdn Bhd do if the contract is frustrated?
5. Does Freedom Flight Sdn Bhd have to return all the money they have received from Monash Soccer Team if the |
| CAM 12 | 1. Does the economic downturn amount to a frustrating event?
2. Does the economic downturn renders performance of contract impossible or radically different?
3. Would Ron Realty Sdn Bhd still be able to rely on doctrine of frustration if the economic downturn made the perf
4. Would  Ron Realty Sdn Bhd still be able to rely on doctrine of frustration if the contract between them contain a |
| CAM 13 | 1. Is Cathy's promise to pay Dave RM500 for not smoking supported by valid considerations
2. Is Cathy's promise to buy Dave a car supported by valid considerations
3. Is Cathy's promise to pay Dave RM1000for all his hard work supported by valid considerations
4. Does the relationship between Cathy and Dave affect the enforceability of the promise to pay RM500, RM 1000
5. Is Cathy's promise to Dave supported by intention to create legal relations? |
| CAM 14 | 1. Is the agreement between Lilian and Mike supported by intention to create legal relations?
2. Did Lilian accept Neil's offer to sell the gold dye at RM 150 each and silver dye at RM 100 each?
3. Does Neil's silence amount to valid acceptance?
4. Is Neil's statement regarding the quality of dye an enforceable term?
5. If Neil's statement regarding quality of dye is not an enforceable term, what can Lilian do?
6. Can Lilian rely on the sale of goods act 1957? |
| CAM 15 | If there is a suggestion of fact which is untrue
If zakariah does not believe suggestion of fact to be true
If Zakariah made statement with intention to induce party to enter into the contract
If consent is caused by fraud |
| CAM 16 | If there is non-disclosure of fact that amount to breach of duty
If non-disclosure of fact induced Dr Pritam to enter into contract
If consent is caused by misrepresentation |
| CAM 17 | 1. If ali and bakar are in a contract
2. If ali's action deprived him from performing his contractual obligations with Bakar
3. If Bakar is a third party to Ali and Chong |
| CAM 18 | 1. If alpha can recover RM 1000 a day
2. If Alpha can recover RM1000 for the entirety of 3 weeks 3. If Alpha can recover RM5000 spent for replacement r
4. If alpha can recover RM 50000
5.If Beta have actual knowledge of RM50000 contract |
| CAM 19 | If party intended for oral statement to have contractual effect as a term of the contract
If party express importance of oral statement
If there is long interval
If oral statement is included in the contract
If oral statement forms collateral contract
If oral statement induce contract
If oral statement is supported by intention
If oral statement is inconsistent with written term
If written term is incorporated by signature
If exceptions to signature rule is applicable |

| | |
|---|---|
| | If party intended for oral statement to have contractual effect as a term of the contract |
| | If party express importance of oral statement |
| | If there is long interval |
| | If oral statement is included in the contract |
| | If oral statement forms collateral contract |
| | If oral statement induce contract |
| | If oral statement is supported by intention |
| | If oral statement is inconsistent with written term |
| | If written term is incorporated by signature |
| CAM 20 | If exceptions to signature rule is applicable |

### A.3.2 Questions for evaluation

- Q1: Is the answer make a correct conclusion?

- Q2: Is there any improvement of the answer? (Only applied for in-context learning and decomposition questions)

- Q3: Is the answer fluency?

- Q4: Is the answer provided information relevance?

- Q5: Is the answer have correct Concept Identification?

- Q6: How many assumptions are made?

- Q7: Among the assumptions made by CHAT-GPT,how many assumptions are correct?

- Q8: Compare the assumptions from CHAT-GPT and humans, how many of them are matched?

- Q9: Please provide your general comment for the given answer.