# OpenReview forum: "Can ChatGPT Perform Reasoning Using the IRAC Method in Analyzing Legal Scenarios Like a Lawyer?"
_EMNLP/2023/Conference — EMNLP 2023 Findings_

### Official Review · Reviewer_Vapu · 2023-07-24

**Soundness:** 2

**Excitement:**

3: Ambivalent: It has merits (e.g., it reports state-of-the-art results, the idea is nice), but there are key weaknesses (e.g., it describes incremental work), and it can significantly benefit from another round of revision. However, I won't object to accepting it if my co-reviewers champion it.

**Paper Topic And Main Contributions:**

The paper proposes a new corpus focused on the study of the capacity of LLMs to carry out legal analysis using the IRAC method. The dataset is composed of 50 scenarios belonging to the Australian Social Act and to the Contract Acts of Malaysia.

In addition, an empirical evaluation is proposed in the aforementioned corpus, using ChatGPT to perform legal analysis using IRAC. This result shows the difficulties that this LLM has to carry out correct reasoning following the legal methodology used.

**Reasons To Accept:**

- It focuses on the generation of resources in a complex research line such as reasoning by language models and legal reasoning.

- A series of experiments are presented to evaluate the feasibility of the proposal.

- A set of guidelines used to create the dataset are provided.

**Reasons To Reject:**

- The proposal is presented as a general study on the legal reasoning capacity of LLMs. However, it is only experienced using ChatGPT.

- Missing is the proposal of a series of hypotheses that are intended to be evaluated in the work, beyond the general study on whether ChatGPT is capable of performing legal reasoning.

- No analysis or insights are presented as to why the results are unsatisfactory and how they could be improved.

- The proposed dataset is not available.

**Reproducibility:**

1: Could not reproduce the results here no matter how hard they tried.

**Reviewer Confidence:**

3: Pretty sure, but there's a chance I missed something. Although I have a good feel for this area in general, I did not carefully check the paper's details, e.g., the math, experimental design, or novelty.

---

> ### Author Rebuttal · Authors · 2023-08-27
>
> Many thanks for your insightful comments and constructive feedback on our work. The following are our responses to the reasons to reject:
>
> R1: Experiments were conducted only using ChatGPT.
> Response: Our work constructs a dataset for IRAC analysis and focuses on understanding to what degree ChatGPT is able to perform reasoning for legal scenario analysis similar to legal professionals. To address this research question, we need to evaluate the best performing LLM for IRAC analysis. If such a model fails to achieve that goal, the other LLMs fail as well. ChatGPT was the best candidate when we started experimenting on the constructed dataset. It performed far better than the other open source LLMs back then. It was the only model that generated meaningful results. The other LLMs failed almost completely, including drawing meaningful conclusions. Given the evaluation cost is high for each model due to the use of legal professionals as annotators, the high cost for the other models would not bring in new information but show that the other models fail almost completely.
>
> R2: Missing a series of hypotheses that are intended to be evaluated in the work.
> Response: As stated in Line 039-042, this work focuses on understanding to what degree ChatGPT is able to perform reasoning for legal scenario analysis in the same way as legal professionals. We have a series of hypotheses to support this research goal, but the hypotheses were not all formulated in the typical formats of hypotheses.
>
> Hypothesis I: The key hypothesis in Line 411 and 447 is that ChatGPT cannot produce the same or correct reasoning steps as those of legal professionals, despite the conclusions may be correct. Our empirical study supports this hypothesis and demonstrates the key challenges for current LLMs.
>
> Hypothesis II: The key hypothesis in Line 449 and 465 is that reasoning paths, especially intermediate reasoning steps, are important for drawing correct conclusions. We support this hypothesis by evaluating ChatGPT with progressively partial reasoning paths as additional inputs. The results show the effectiveness of reasoning paths for final conclusions. This also sheds light on how future models should improve. Not just predicting the final conclusions but also focusing on generating correct intermediate reasoning steps.
> Hypothesis III: Line 466 - 489 are concerned with the hypothesis that similar to humans, demonstration with reasoning examples is useful for improving reasoning performance. In the corresponding experiments, we show that in-context learning with the reasoning examples similar to the input scenario is effective in improving reasoning performance. This motivates future research on how to build and retrieve similar examples for improving reasoning.
> Hypothesis IV: Line 490 - 515 focus on the hypothesis that a key factor of poor performance is that the initial question is too complex for ChatGPT to answer. Then we show that when decomposing a legal question into simpler questions and let ChatGPT answer them respectively, the performance is greatly improved.
>
> We will revise the corresponding sections to make hypotheses, findings, and support of hypotheses more clear.
>
> R3: No analysis or insights are presented as to why the results are unsatisfactory and how they could be improved.
> Response: Line 434 - 447, we show that ChatGPT produces unsatisfactory analysis parts because i) most of the intermediate reasoning steps fail to include correct references to statutes and precedents; ii) the articulation of reasoning in the analysis parts is sometimes confusing and logically inconsistent; iii) most of the analysis parts from ChatGPT do not align with those from legal professionals.
> Line 448 - 465 shows that the above limitations can be mitigated by feeding partial reasoning paths written by professionals to ChatGPT. It sheds light on the research direction about how to overcome the problems. Moreover, Line 466 - 489 shows the research direction how to further improve the reasoning performance by providing similar in-context examples.
> Line 490 - 515 shows that one of the reasons for unsatisfactory performance is the complexity of legal questions. If we decompose the question into simpler ones, the performance will be greatly improved. The future research direction is to automatically generate decomposing questions for complex legal questions.
>
> R4: The proposed dataset is not available.
> The proposed dataset is provided in the submitted supplementary materials. However, we got to know after the submission that the supplementary materials may not visible to reviewers.

---

### Official Review · Reviewer_623p · 2023-08-03

**Soundness:** 5

**Excitement:**

4: Strong: This paper deepens the understanding of some phenomenon or lowers the barriers to an existing research direction.

**Paper Topic And Main Contributions:**

The paper conducts an analysis of the legal reasoning capabilities of ChatGPT on a range of legal problems, creating at the same time an annotated dataset for legal reasoning on a sample from Contracts acts from Malaysia and the Australian social act for dependent children. The authors demonstrate 1/ the misalignment of the "reasoning" steps of ChatGPT compare to a legal professional and 2/ the lack of autonomy of ChatGPT when reasoning about a legal case, that improves as the prompts get more detailed.

**Questions For The Authors:**

- The evaluation is conducted on 2 very small areas of law, how generalizable could those results be? It seems that those areas are also quite "easy" areas compared to some more potentially complex legal domains that rely on more statutes/ more precedents?

**Reasons To Accept:**

- The topic of the study is very relevant and should be of interest not only to the NLP community interested in domain-specific applications, but also to legal professionals or scholars.
- The methodology,  implementation and evaluation are very clearly and precisely explained (both in the main paper and in annex), from the used legal cases, the annotators recruitment and level of expertise, the questions asked etc. This is highly appreciable, helps the reader to follow and demonstrate the quality of the survey.
- Legal NLP generally lacks resources and datasets, and for this reason, introducing this new dataset in English is important. Moreover, this dataset is a valuable contribution on 2 aspects: 1/ it includes scenarios from Australia and Malaysia which are two under-represent jurisdictions in the existing legal NLP English resources. 2/ It provides human annotations on legal scenarios in a very precise manner including the legal reasoning that is thought as a demonstration and justification, which is very valuable for a variety of NLP tasks including legal QA.
- The assumption made line 327 that the desired output is aligned with the human reasoning behaviour is important.
- The results provide a good foundation for further research in legal QA/generative NLP.

**Reasons To Reject:**

- line 449 "Impact of adding reasoning paths": this experiment yields important results, however, one could wonder if providing too much information to ChatGPT is realistic? It seems natural that the more information you provide, the better it performs. However the goal of using a QA system in the legal domain is rather to ask a simple question and get an answer, rather than already providing a reasoning path.
- The idea of introducing a neuro-symbolic approach and logical operators is interesting. Similarly to the above comment, I wonder if it wouldn't be more meaningful and realistic to consider only natural language, for purposes of simplicity, wider understanding including by the users and because having to re-write a reasoning process takes time. This may be counterproductive.

**Reproducibility:**

4: Could mostly reproduce the results, but there may be some variation because of sample variance or minor variations in their interpretation of the protocol or method.

**Reviewer Confidence:**

5: Positive that my evaluation is correct. I read the paper very carefully and I am very familiar with related work.

**Typos Grammar Style And Presentation Improvements:**

- It would be beneficial to explain earlier in the paper what is meant by "semi-structured language", for example in introduction. The reader only understands that key element of the study around line 216 when it has been mentioned several times before.
- Section 3.4: the compute of the "average length" is not clear. The reader understands that it is a good proxy for complexity, one would like to understand what is considered a "reasoning step".
- Section 5.1: there are a few spaces missing in the citations: line 551 "Andew(Bl...", line 552, line 575

---

> ### Author Rebuttal · Authors · 2023-08-27
>
> We greatly appreciate your insightful feedback and comments. Below are our responses to the reasons to reject.
>
> R1: Impact of adding reasoning paths.
> Response: The purpose of our experiments on feeding progressively partial reasoning paths to ChatGPT are two-folds. First, the previous subsection shows that the reasoning paths produced by ChatGPT did not align with those of legal professionals in general. We show that providing correct reasoning paths, even part of them, indeed help ChatGPT to improve reasoning performance. Particularly, wrong reasoning paths are the key to poor performance of ChatGPT. Hence, the future research direction is to generate higher quality reasoning paths for both performance improvement and explanabilty. Second, the experiments do not aim to suggest that people should provide correct reasoning paths when using ChatGPT for legal analysis. Those experiments were carefully designed and carried out to demonstrate to what extent correct reasoning paths are useful. E.g. what if 20%, 40% or 80% of them are correct. How would correct reasoning paths influence final conclusions? We will revise this paragraph to make our purpose clearer.
>
> R2: usefulness of a neuro-symbolic approach for legal reasoning.
> Response: The neuro-symbolic approach aims to make the reasoning of LLMs more robust and generalizable across different scenarios. The recent evaluations of LLMs show that LLMs cannot work well in several tasks, e.g. numerical reasoning, complex reasoning, and compositional generalization. Unfortunately, legal reasoning indeed associates with some of the key weaknesses of LLMs. To address those weaknesses, we introduce symbols only at the places where LLMs often fail. When building a stronger LLM, such symbols will be associated with the corresponding functions and use of symbols will support advanced (symbolic) reasoning techniques, which will produce more reliable results than using language alone. Moreover, the language used in statutes already use the same words for the same logical relations. Hence, we only need to associate such keywords with the corresponding operators without rewriting the reasoning processes. In the legal domain, some scholars are trying similar ideas by making legislations more structured, even though they do not have a computer science background.
>
> Answers to the questions for the authors:
>
> Q1: Small areas of law
> Answer: The reasons that our findings are generalizable to many areas of law are three-folds. First, our dataset and evaluation covers all four aspects of IRAC analysis for each scenario. Second, the scenarios in Contract Law are from tutorials and exams for law students, they cover all main topics in that area, so that they are not simple. Law students often need several hours to analyze them. The running examples in the main body of the paper are chosen due to the limited space. Third, the evaluation we conducted does not involve any assumptions or heuristic rules specific to the two selected areas. The more complex scenarios can be considered as having more statutes and precedents involved. Compared to the existing datasets on legal reasoning, such as SARA (Holzenberger, N., & Van Durme, B. (2021), our datasets are far more comprehensive and closer to real-world scenarios than those datasets.
> For the typos and grammar style and presentation improvements:
>
> 1: Introducing semi-structured language earlier.
> Answer: We will revise this manuscript to introduce semi-structured language in Introduction.
>
> 2: What is the “average length”?
> Answer: The length of a reasoning path is defined as the number of involved reasoning steps. A reasoning step involves the application of a single rule or a single argument from a precedent to draw conclusions. The average length is thus the average number of reasoning steps across the reasoning paths of interest.
>
> 3. Missing spacing for the reference
> Answer: We will revise the spacing in line 551.

---

### Official Review · Reviewer_uuNg · 2023-08-05

**Soundness:** 2

**Excitement:**

3: Ambivalent: It has merits (e.g., it reports state-of-the-art results, the idea is nice), but there are key weaknesses (e.g., it describes incremental work), and it can significantly benefit from another round of revision. However, I won't object to accepting it if my co-reviewers champion it.

**Paper Topic And Main Contributions:**

The paper explores the application of Large Language Models (LLMs), specifically ChatGPT, in the legal domain for performing legal analysis using the IRAC (Issue, Rule, Application, Conclusion) method. To assess ChatGPT's ability to reason like legal professionals, the authors construct a novel dataset called SIRAC, containing legal scenarios related to the Contract Acts of Malaysia and the Australian Social Act for Dependent Children. Each scenario is annotated with a complete IRAC analysis in a semi-structured format. The experimental results reveal that while ChatGPT can produce reasonable answers, it struggles to generate correct reasoning paths aligned with legal experts. The paper highlights potential research directions to enhance the alignment between LLMs and legal professionals in legal reasoning tasks.

**Reasons To Accept:**

(1) Novel Corpus Construction: The paper presents the creation of a novel dataset, SIRAC, containing annotated legal scenarios, which not only aids legal professionals but also serves as a valuable resource for legal Natural Language Processing (NLP) research.

(2) Evaluation of ChatGPT for IRAC Analysis: The paper provides the first empirical assessment of ChatGPT's performance in legal reasoning using the IRAC method. This evaluation sheds light on the strengths and limitations of LLMs in legal tasks.


**Reasons To Reject:**

(1) Lack of Related Work on IRAC Methodology: The paper introduces the IRAC methodology for legal reasoning (Alsagoff, 1996), but it lacks a comprehensive review of related work on this specific method. Including a broader discussion of previous research and approaches using IRAC would provide a more thorough understanding of its effectiveness and limitations in legal analysis.

(2) Limited Contribution to Specific Countries: The paper focuses on legal scenarios from Malaysia and Australia, which may limit the generalizability of the findings to other legal systems and jurisdictions. Expanding the dataset to include scenarios from a more diverse set of countries would enhance the applicability and relevance of the study.

(3) Small Dataset: The SIRAC dataset contains only 50 legal scenarios related to ASA and CAM. This limited dataset size may impact the robustness and generalizability of the results. A larger and more diverse dataset would strengthen the validity of the findings and improve the model's performance on various legal scenarios.

**Reproducibility:**

N/A: Doesn't apply, since the paper does not include empirical results.

**Reviewer Confidence:**

4: Quite sure. I tried to check the important points carefully. It's unlikely, though conceivable, that I missed something that should affect my ratings.

---

> ### Author Rebuttal · Authors · 2023-08-27
>
> Many thanks for your helpful comments on our work. The following are our responses to each reason to reject:
>
> R1:  Lack of comprehensive review of IRAC Methodology.
> Response: We introduce IRAC in Sec. 2, which should provide enough information for understanding our work. In addition, we discuss the relevant works we are aware of, which use LLMs for IRAC analysis in Sec. 5.1. Herein, (Yu et al., 2022) covers only the conclusion parts of IRAC and (Guha et al., 2022) divide IRAC into independent simplified tasks and each task is applied to different simplified scenarios. Thus they do not provide comprehensive evaluation from issues to conclusions on the same scenarios. In contrast, our work is the most comprehensive and complete evaluation of an LLM for IRAC analysis on legal scenarios, and provides the first dataset for such an evaluation. Moreover, IRAC is the most widely used legal reasoning framework for legal professionals. A comprehensive review of this technology is conducted often in textbooks and long surveys for law students and junior legal professionals. Therefore, we believe a comprehensive review of IRAC goes beyond the scope of this work.
>
> R2:  Limited Contribution to Specific Countries.
> Response: Legal reasoning is challenging even for legal professionals. Therefore, only top law students, lawyers and law professors are able to help with annotations and evaluation of the IRAC methodology applied to scenarios. The analysis of legal scenarios is expensive and time consuming, which usually takes more than two hours per case. The evaluation of legal analysis even takes significantly more time. The Jurisdictions and the legal systems vary significantly so that legal professionals in one legal system cannot work on the other legal systems. As a result, our annotators cannot work on the legal scenarios from the other countries. We believe, finding top law scholars and top law students in foreign countries and having the capability to pay them remotely is beyond the capabilities of most research labs in this field. Our strategy is to start with the common law system, and choose the countries that it is unlikely that LLMs already memorize a large number of similar cases and relevant statutes so that the legal reasoning is not simplified to a retrieval problem. We identify the challenges and open problems by using the dataset we constructed and claim the findings in the common law system. We hope, our work will promote similar research in other jurisdictions and countries. In our point of view, although it is desirable to build a dataset and conduct evaluation across all jurisdictions and legal systems, it needs the efforts from the community, not just a single research lab. Similar to the first work on machine translation in our community, which just translated 60 Russian sentences into English, our work kick-starts a new line of research and shows that current LLMs fail to achieve reasonable performance on such tasks.
>
> R3: Small Dataset.
> Response: As stated in Line 039-042, this work focuses on understanding to what degree ChatGPT is able to perform reasoning for legal scenario analysis in the same way as legal professionals. The key finding is that on all scenarios analyzed by ChatGPT, most of the intermediate reasoning steps in the analysis parts are wrong, though some of the conclusions are correct. None of the scenarios has a completely correct analysis part. When we feed correct human-written partial reasoning paths to ChatGPT, the reasoning performance of ChatGPT is improved significantly and consistently. Hence, this key finding is not based on selection bias or randomness, and the finding is adjusted by the corresponding interventions (providing partial reasoning paths). The same is applied to the other findings. In other words, the dataset is large enough for the findings identified in our work. We agree that it is always better to have larger datasets for the community. However, this task is special due to the high requirements on financial resources and annotators. If we want the dataset to have similar size as the other NLP tasks, probably only the richest labs in the world are able to conduct this line of research. The outcomes of this line of research are for social good. Hence, we believe it is better off encouraging more research teams to participate in this line of research, as long as their findings can be well adjusted by their empirical evidence.

---

### Meta-Review · Area_Chair_R3w5 · 2023-09-16

**Recommendation:** 3

**Metareview:**

The paper aims at assessing the legal reasoning capabilities of ChatGPT and introduces a novel dataset annotated on the bases of the IRAC methods. The experimental results reported offer useful insights into the limitations of current technology. While one reviewer was concerned about the limited coverage of the data set to only two countries and a few cases, it still offers an interesting example for future research. However, the paper needs improvements.

**Pros**

- The dataset is valuable and has the potential to help research on legal NLU;

- the topic is timely and highly relevant not only for NLP

- the paper is clear and easy to follow;

- evaluation experiments are adequate and methodologically sound, although not sufficient to generalize the claims to other LLMs;

**Cons**

- the discussion of related works could better motivate the soundness of the IRAC methods as guiding principles for analysis;

- the coverage of legal cases is limited, albeit perhaps sufficient to support the claims of this paper. A critical comparison with the other larger datasets mentioned by reviewers would be beneficial;

- the experimental set up should be better motivated and explained, to avoid misunderstandings;

---

### Decision · Program_Chairs · 2023-10-07

**Decision:**

Accept-Findings

**Comment:**

The paper aims at assessing the legal reasoning capabilities of ChatGPT and introduces a novel dataset annotated on the bases of the IRAC methods. The experimental results reported offer useful insights into the limitations of current technology. While one reviewer was concerned about the limited coverage of the data set to only two countries and a few cases, it still offers an interesting example for future research. However, the paper needs improvements.

**Pros**

- The dataset is valuable and has the potential to help research on legal NLU;

- the topic is timely and highly relevant not only for NLP

- the paper is clear and easy to follow;

- evaluation experiments are adequate and methodologically sound, although not sufficient to generalize the claims to other LLMs;

**Cons**

- the discussion of related works could better motivate the soundness of the IRAC methods as guiding principles for analysis;

- the coverage of legal cases is limited, albeit perhaps sufficient to support the claims of this paper. A critical comparison with the other larger datasets mentioned by reviewers would be beneficial;

- the experimental set up should be better motivated and explained, to avoid misunderstandings;